:ᐳᕀ: PLOS | ONE

# The effect of various metal-salts on the sedimentation of soil in a water-based suspension

**Andras Sebok**[1]*, **Viktoria Labancz**[1], **Imre Czinkota**[1], **Attila Nemes**[2]

**1** Department of Soil Science and Agrochemistry, Szent Istvan University, Godollo, Hungary, **2** Norwegian Institute of Bioeconomy Research, As, Norway

\* sebok.andras@mkk.szie.hu

**Data Availability Statement:** All relevant data are within the paper and its Supporting Information files.

## Abstract

Soil particles and bound nutrients that erode from agricultural land may end up in surface waters and cause undesirable changes to the environment. Various measures, among them constructed wetlands have been proposed as mitigation, but their efficiency varies greatly. This work was motivated by the assumption that the induced coagulation of particles may accelerate sedimentation in such wetlands and by that help reduce the amount of material that is lost from the vicinity of the diffuse source. Our specific aim was to laboratory-test the effectiveness of various salt-based coagulants in accelerating the process of sedimentation. We tested the effect of $Na^+$, $Mg^{2+}$, $Ca^{2+}$, $Fe^{3+}$ and $Al^{3+}$ cations in 10, 20, 40 and 80 mg $L^{-1}$ doses added to a soil solution in select, soluble forms of their chlorides, sulphates and hydroxides. We mixed such salts with 1 gram of physically dispersed, clay and silt rich (>85% in total) soil material in 500 mL of solution and used time-lapse photography and image analysis to evaluate the progress of sedimentation over 3 hours. We found that 20–40 mg $L^{-1}$ doses of $Mg^{2+}$, $Ca^{2+}$ in their chloride or sulphate forms appeared to provide the best consensus in terms of efficiently accelerating sedimentation using environmentally present and acceptable salts but keeping their dosage to a minimum. Comprehensive in-field efficiency and environmental acceptability testing is warranted prior to any practical implementation, as well as an assessment of small scale economic and large-scale environmental benefits by retaining soil and nutrients at/near the farm.

## Introduction

The loss of significant amounts of soil particles and valuable nutrients from land-based agriculture due to erosion processes is a global problem and is among the primary polluters to surface waters in Norway [1]. It is an economic and sustainability issue for farmers, and the water quality issues downstream are a concern to local and national governments and to the general public. Sediment and nutrients that are not captured close to their source will eventually end up in large surface water bodies and eventually in the sea, and it will take an insurmountable cost to recover any of it for active land-based use.

**Funding:** The measurements were supported by the Hungarian EFOP-3.6.1-16-2016-00016 programme.

**Competing interests:** The authors have declared that no competing interests exist.

In order to help reduce such impact downstream, sedimentation ponds–as part of constructed wetland systems–are increasingly coupled with in-field measures on agricultural land. Such ponds carry great potential towards intercepting and potentially recycling soil material and bound nutrients. Ponds being installed in catchments in order to help retain and reuse solids and nutrients from discharged water have been reported by e.g. [2] and [3]. The efficiency of sedimentation ponds can vary greatly; literature cites sediment type, the pond's dimensions and its nominal hydraulic loading as most influential factors that determine their efficiency ([4], [5]). Weather patterns may also strongly influence their performance due to variability in the intensity and duration of rainfall events, and the potential hydraulic overloading that is caused. When intensive rain events occur, the hydraulic retention time of a pond decreases, which can lead to reduced sedimentation efficiency. The potentially increasing frequency of intensive rain events associated with future climate predictions may make the prediction of pond efficiency be even less certain in the future, because water-erosion processes and resulting sediment loading are closely related to rainfall-runoff intensity [6].

Nutrients tend to bind in greater relative amounts to smaller particles due to their greater relative surface area and by that their greater adsorption capacity ([7], [8]). A study reported that more than 50% of phosphorus was bound to the fraction that was smaller than sand [9]. At the same time, it is of particular challenge to sediment the smaller, silt or clay sized particles as their suspension flows through a pond(-system). Particle-size in the clay fraction is close to or already inside the colloid size-range, which have practically no settling ability (few cm/day, [10]); and these particles may remain suspended in the suspension by Brownian motion. The combination of being rich in nutrients and being difficult to sediment requires special focus on the clay and silt fractions in efforts to improve the quality of surface waters.

According to [4], increased aggregation of the suspended solids could yield more effective settling and reduce the progression of sediment and nutrients downstream. We therefore initiated a study to find potentially effective, cheap and environmentally friendly coagulants that may be usable to enhance sedimentation in-situ. As several authors pointed out before (e.g. [11], [12], [13]) cations tend to help aggregate soil particles in a suspension ([14, 15]). Suspended solids have a negative surface charge that prevents them from aggregating into bigger particles. The mechanism behind the proposed approach is that this double layer (Stern-double layer or diffuse double layer; repulsive force between colloids–[16]) could be decreased if positively charged cations are added to the system. This is a well-known phenomenon that is used as part of wastewater treatment technologies [17], where coagulants are used widely to settle the smaller (colloid size) particles. Different cations have different potential to enhance such coagulation. The Shulze-Hardy rule says that an increase in the charge of the applied cation has a power-law relationship with the cation's ability to enhance coagulation [18]. In field conditions, this coagulation, however, is reversible, in case the solution is diluted. The stability of aggregates may also decrease, as bonding by organic compounds and iron/manganese/calcium bonds between particles weaken and may eventually disappear over time [19]. This is a condition to be avoided, as an increase in the amount of smaller particles may enhance nutrient-release [20], [21]. Contrary to cations, the message about anions in the literature is much less clear. For example, some authors described that anions, especially sulphate has positive effect on the sedimentation process [22]. On the contrary, also a negative effect of sulphate has been reported [23].

Our working hypothesis is that it is possible to find metal-salt or salts that can be used in-situ to enhance particle coagulation and by that improve their sedimentation in open water bodies, such as e.g. sedimentation ponds, in an economically feasible and environmentally acceptable manner. In order to reduce the impact on the environment, in this paper we report on a laboratory experiment in which we examined the effectiveness of a matrix of cation-anion

pairs on sedimentation. Feasibility and biological acceptability are studied in follow-up field-based research with only select salts.

## Materials and methods

### Sedimented material

For the experiment we used soil material from the Skuterud field-site near Ås in SE Norway (location, DMS: Latitude N 59˚ 40' 12.652", Longitude E 10˚ 50' 15.904", Altitude 119 m MASL), collected within the framework of the Soilspace Project (NFR Project #240663, Norway). The soil is a Gleyic Stagnosol that was formed from marine sediment, and represents the type of soil material that often erodes into waterbodies in the region. The sample used throughout the experiment originates from the 70-75cm depth with low (0.17%) organic matter content and was only physically crushed and homogenized. No further treatment was applied in order to preserve the chemical composition of the soil. This soil mix contained 13.9% clay (< 2μm), 73.1% silt (2–50 μm), 10.7% very find sand (50–100 μm), 2.4% fine sand (100–250 μm), and no medium or coarse sand according to the USDA [24] system (Fig 1). The effective size distribution was determined by a laser diffractometer (Malvern Mastersizer 3000 –Hydro LV), but we note that this size distribution has only functional meaning for this study and is not equivalent to this soil's particle size distribution, due to deviations from protocol in the sample pre-treatment. In particular, we did not use Tetrasodium pyrophosphate ($Na_4P_2O_7$), the chemical dispersant often prescribed in particle-size measurement protocols.

### Tested ion combinations

The combinations of anions and cations used in this study are depicted in Table 1. We tested 10, 20, 40 and 80 mg $L^{-1}$ concentrations of the respective cations in the solutions. These were easily accessible salts and were not deemed harmful to the environment in the short run. From Fe and Al salts we only tested one to test and prove the concept, knowing that their open-water applicability in practice is unlikely due to longer term environmental considerations and existing policy on environmental release (EU Water Framework Directive, [25]). We elected to work with Fe and Al sulphates only since their chlorides and hydroxides are not commonly used in water treatment processes.

### Experimental setup

We mixed 1.0 gram of soil and 500 mL of distilled water in standard, transparent laboratory glass cylinders. The sediment concentration was chosen to represent, but somewhat exceed, historic high concentrations of total suspended solids found in runoff water in boreal environments (1–1.3 g $L^{-1}$, [26]). Five sedimentation cylinders were tested in parallel, each round of sedimentation containing the 4 concentration levels of the same cation/anion combinations (for example, $CaCl_2$, Fig 2) and a control tube with no salt added to it. We constructed and tested a chamber containing a pair of line-LED light-sources in a horizontal configuration and curved matte-white reflection background that scattered light and provided stable and constant indirect back-lighting to the cylinders. Still, the results of every run were normalized to their own control cylinder in order to remove any potential variation in light conditions caused by e.g. unexpected fluctuations in power, the potentially different white balance caused by the camera, or other factors that could potentially have differed between runs. The JPEG pictures were taken by a Fujifilm FinePix S8600 camera in manual mode, with settings of 4608 x 2592 pixels (16:9, L, fine quality), shutter speed 60 and aperture F2.9. We used an electric stirrer to stir the contents of the cylinders for 30 seconds, took three pictures of the set of five

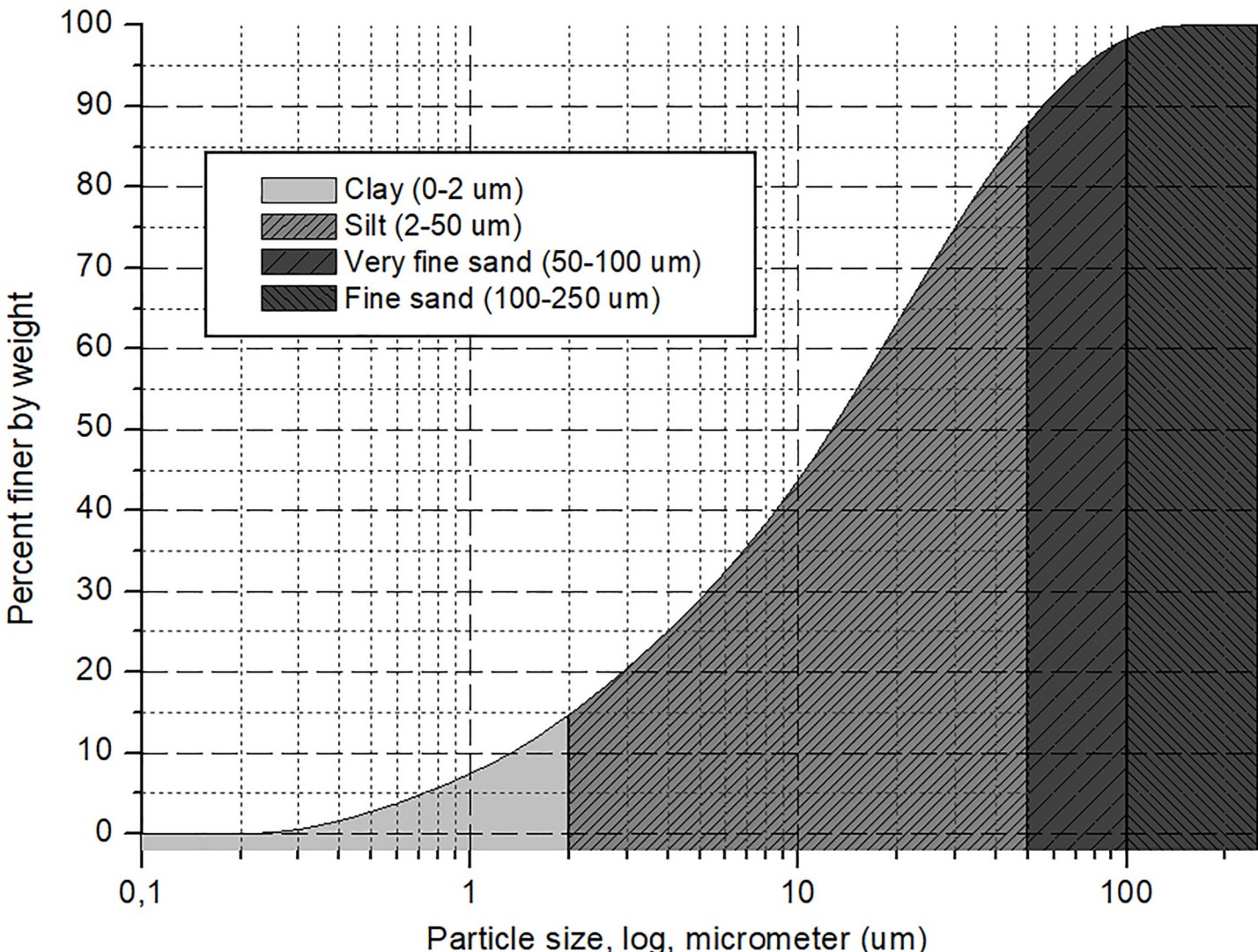

**Fig 1. Size distribution of the sedimented soil material, determined by laser diffractometry.** Note: The soil was not submitted to chemical pre-treatment for disaggregation.

cylinders every 10 minutes from minute 0 to 180, and chose the sharpest of the three pictures to analyse. The camera and the cylinders were replaced to marked positions and orientation for every run. The pH of each solution was tested at the beginning of the experiment upon

**Table 1. Salts used as coagulants during the experiment.**

|  | $Cl^-$ | $SO_4^{2-}$ | $OH^-$ |
|---|---|---|---|
| $Na^+$ | ✓ | ✓ | ✓ |
| $Mg^{2+}$ | ✓ | ✓ | Not soluable |
| $Ca^{2+}$ | ✓ | ✓ | ✓ |
| $Fe^{3+}$ | Not tested | ✓ | Not tested |
| $Al^{3+}$ | Not tested | ✓ | Not tested |

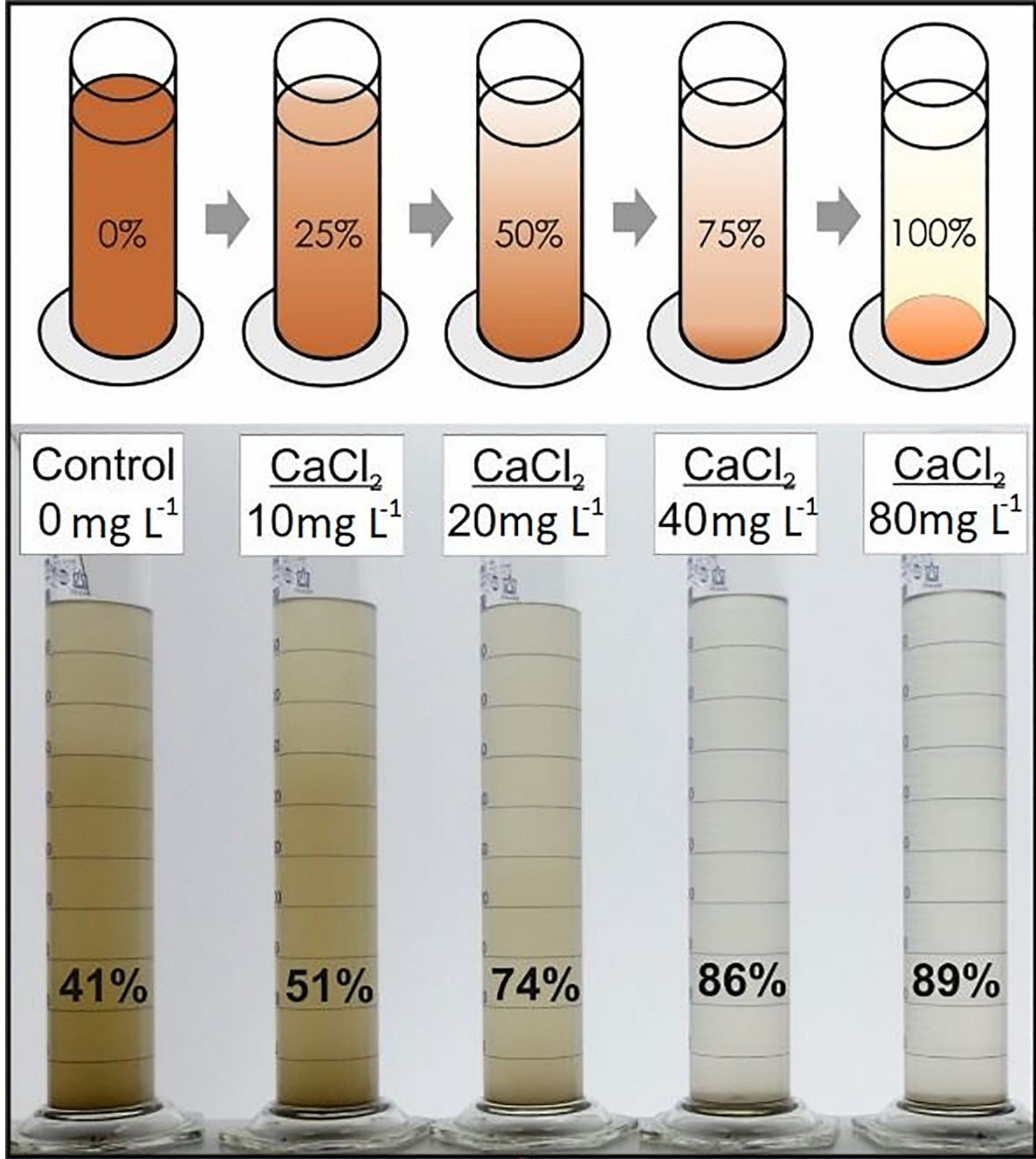

**Fig 2. Top: Concept of differences in turbidity being an indicator of progress in sedimentation.** Bottom: glass cylinders with 4 ion application rates of Ca in the form of CaCl₂ and an ion-less control, after 3 hours.

mixing the suspension, as well as after the 3-hour experiment using a Meinsberg TM40 portable pH meter.

## Image analysis and interpretation

It is a reasonable assumption that the turbidity in a sedimentation cylinder relates to the amount of sedimented material [27], [28]. We developed task-specific software in-house in the Delphi developer environment to determine the average colour within each cylinder.

Horizontally, we selected a 20% subset of the front of the cylinder (35 pixels wide) to avoid any effects from distortions and any light reflection due to the round shape of the cylinders. Vertically, we generated the average RGB value across a 400-pixel zone-of-interest–which represented the middle third of the solution-filled portion of the cylinders—to smooth out any pixelation. We note that image properties obtained in this study were not directly associated with levels of suspended sediment content using conventional laboratory methods. The transparency of a cylinder containing only distilled water was also measured and was used as reference to a 100% clean system in terms of sedimentation efficiency. In each run, the colour of the freshly stirred (0 minute) solution was used as reference to 0% efficiency of sedimentation in order to provide a closed (0–100%) relative linear scale for each single run (according to [28]). We acknowledge that despite every effort to obtain a fully homogeneous back-light, it showed a small, but measurable unevenness. We used the experimental step to correct for potential differences between runs; and used the image-processing step to account for any resulting differences within a run. We used a vertically parallel portion of the background adjacent to each cylinder as a correction factor to any variability both in the horizontal and vertical direction.

The time-series of images taken over 180 minutes were used to account for the progression of sedimentation, expressed within the 0–100% scale. Upon fitting an exponential curve to the obtained data, we obtained the expected end point in the degree of sedimentation in response to each ion-concentration treatment and used the t value–multiplied by ln2 –to determine the time needed to reach 50% of the expected end point (>99%) of sedimentation (later termed 'half-time') as an indication of the speed of sedimentation. We note that this t value should not be mixed with that of the statistical test.

## Results

Fig 3 and Table 2 depict the obtained data and fitted exponential models for four concentrations of one cation-anion pair, $MgCl_2$, as an example of ion-pairs, plus its control. The plots depict a relatively quick initial settling that slows down and eventually barely or does not continue after some time–as interpreted from the increasing transparency of the suspension using

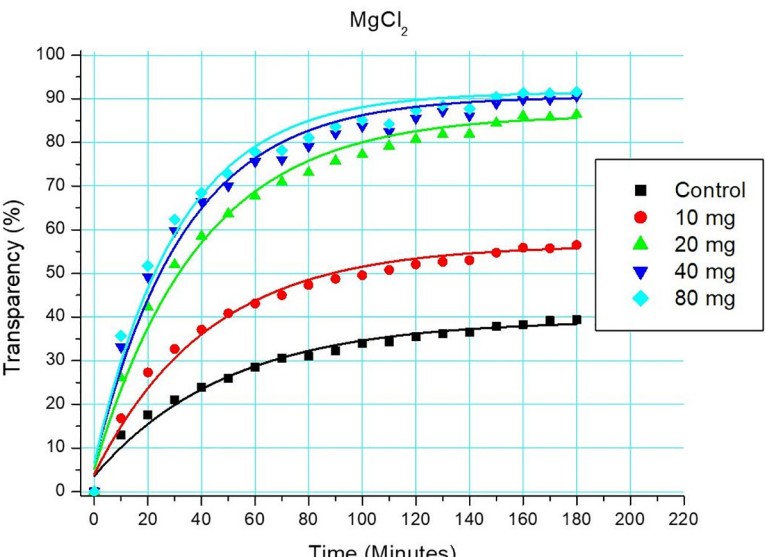

**Fig 3. Sedimentation results of four concentrations of $MgCl_2$ solutions, and a control using only distilled water.**
Exponential curves were fitted using the Origin 6.0 software; curve parameters are presented in Table 2.

Table 2. Curve parameters for the four concentrations of MgCl$_2$ solutions.

| mg L$^{-1}$ cations (Mg$^{2+}$) | Chi$^2$ | R$^2$ | t$_{1,2}$ (mins) |
|---|---|---|---|
| 0 | 0.00021 | 0.981 | 49.4 |
| 10 | 0.00034 | 0.986 | 42.4 |
| 20 | 0.00066 | 0.988 | 39.3 |
| 40 | 0.00117 | 0.979 | 33.7 |
| 80 | 0.00125 | 0.978 | 31.3 |

image analysis. Greater concentrations of the salt lead to greater overall efficiency in helping the sedimentation of solids. For MgCl$_2$, using a concentration of 80 mg L$^{-1}$ of Mg yielded only marginal improvement in terms of the dynamics and overall efficiency of sedimentation over the 40 mg L$^{-1}$ solution. As applied salt concentrations decreased, the differences in sedimentation efficiency became more noticeable. The visual assessment of results is somewhat more difficult in terms of the time needed to reach half the overall efficiency. It is comprehensible, however, that already after ca. 20–30 minutes the 20-40-80 mg treatments reached half their sedimentation efficiency, where more suspended material had settled than the total sedimentation during the 3-hour experiment in the control treatment.

Figs 4–6 and Table 3 allow a quantitative oversight of the two examined metrics—the relative efficiency of sediment-removal and an indicator of the speed of removal later cited as '*half-time*' of the sedimentation process according to first order kinetics, where t$_{1,2}$ = ln2/k, and k is the fitting parameter of the first order kinetic equation. At the examined range of concentrations, salts of the examined monovalent cation (sodium) removed only marginally more sediment from the test solution over 3 hours than the control treatment did. Sodium-hydroxide showed the weakest potential towards enhancing sedimentation.

We recorded pH at the beginning of the experiment after just adding the soil material to the ion-holding water or the control, as well as after the 3-hour experiment concluded. We note that the pH and its change should be looked at in the context of the 'control' (black, labelled diamond) presented for each ion-pair, as there were daily variations in the pH of the available water. Fig 4 shows that the tested hydroxides in general (NaOH, Ca(OH)$_2$) tended to make the solution strongly alkalic, while Na$^+$ as a cation tended to make it slightly alkalic. The tested Al and Fe sulphates clearly yielded gradually more acidic pH with growing ion concentration from the start, contrary to their controls. Calcium chloride and sulphate application seemed to influence pH the least, irrespective of the applied concentrations. Magnesium chloride and sulphate applications started from slightly acidic pH that tended to increase towards neutral pH during the 3-hour experiment. The reason for the noted increase in pH during the application of MgCl$_2$ and MgSO$_4$ is unknown, but in the context of its control run, it does not appear to be due to the application of the salt itself, since the controls presented similar increase. We also measured the redox potential in the solutions, but probably partially due to the short duration of the experiments we did not find informative trends in them.

We performed analysis of variance on the obtained data (S1 Appendix) that showed that the type of cations (Type III Sum of Squares analysis—probability of exceeding F_crit <0.0001) and concentration (<0.0001) are significant factors that determine the overall sedimentation efficiency within 3 hours, whereas anion type was not significant (0.501). Salts of bivalent and trivalent cations were significantly more effective in enhancing sedimentation–in accordance with the Shulze-Hardy law [16] (t <0.0001). Within the examined range of concentrations, the overall efficiency of those salts did not differ very significantly (82–91%). However, it made a difference at what concentration those values were attained. Trivalent (Fe and

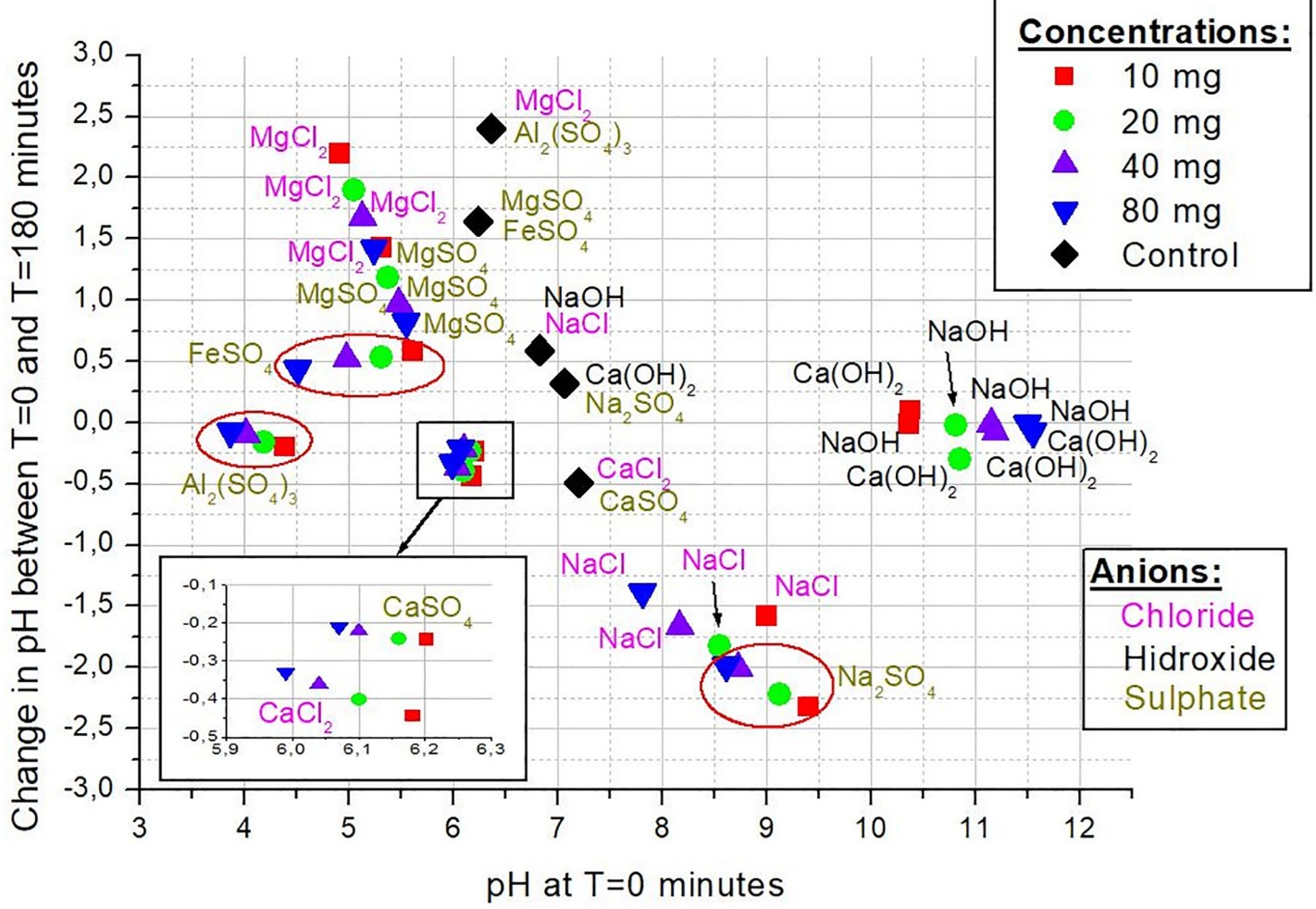

**Fig 4. pH of the suspension upon start of the experiment (i.e. after mixing the soil and applied salts) vs. the change in pH after 3 hours**

Al) salts needed only 10 mg L$^{-1}$ cation concentration to reach that efficiency, while bivalent salts needed 20–40 mg L$^{-1}$. Increasing their concentration, however, did not have further positive impact on sedimentation. The efficiency of Mg and Ca chlorides and sulphates were rather comparable, with the Mg salts being somewhat more effective at 20 mg L$^{-1}$. Visually, we found a weak pattern between the two anions, SO$_4^{2-}$ being somewhat less effective than Cl$^-$ at 20 mg L$^{-1}$ equivalent of the bivalent cation, but not overall. The differences in efficiency of Mg and Ca were tested statistically against the other with the same anion and were found not to differ (S1 Appendix: two sample t-test with equal variances, t = 0.26 and -0.01, t_crit = 2.30), whereas none of the anions showed any better effect than the other. However, literature is inconsistent about the anions' efficiency, and therefore we also caution against drawing wide conclusions about the performance of the examined anions.

In order to demonstrate the repeatability of our experimental results, we triplicated the experiment with the chlorides and sulphates of the bivalent cations. The metric of their final efficiency was rather consistent between repetitions, with only the sulphates presenting somewhat greater variability at 10–20 mg L$^{-1}$ cation concentration (Table 3).

The speed of sedimentation may be important in high(er)-flow situations; therefore we examined an indicator of the speed of sedimentation that we defined earlier (c.f. 'half-time').

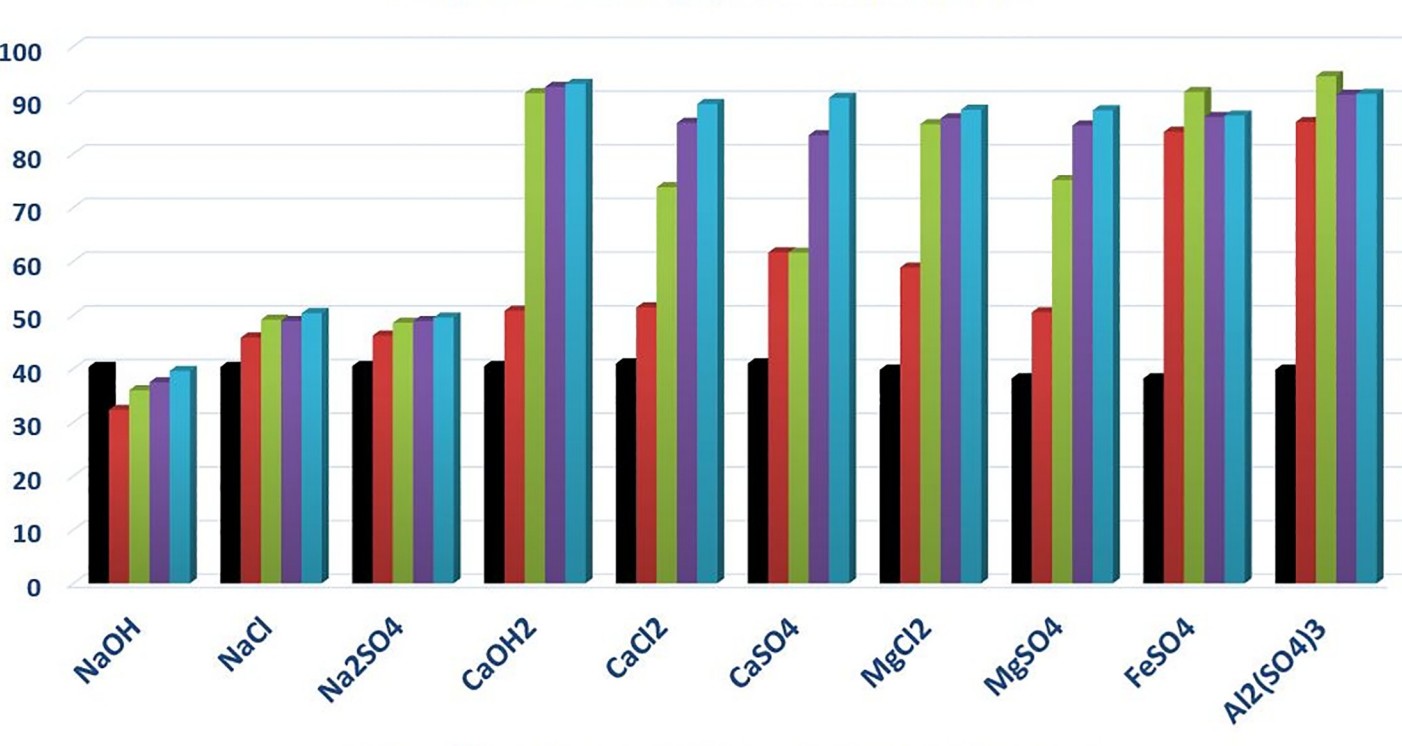

**Fig 5. Overall efficiency of sedimentation after 3 hours, using the cation-anion pairs introduced in Table 1.**

Fig 6 shows decreasing trends with increasing concentrations in general, meaning a faster dynamics to initial sedimentation. Chlorides appear somewhat, faster acting than their $SO_4^{2-}$ counterparts, and a general order of monovalent > bivalent > trivalent salts' half-time is visible, although they generally all fall into the 20–35 minutes range. Triplication of the experiment yielded rather consistent values of the half-times for the treatments (bivalent chlorides shown in Fig 6) but presented somewhat greater variation in the controls (Table 3).

Analysis of variance (S1 Appendix) indicated that the concentration and cation effects are strongly significant (Type III Sum of Squares analysis—probability of exceeding F_crit <0.0001) and contrary to the results obtained about the overall sedimentation efficiency, anions also showed a weak, but significant signal regarding the speed of reaching 'half time' (prob. F = 0.032). Specifically, $Cl^-$ enhanced the speed of sedimentation somewhat over $SO_4^{2-}$ and $OH^-$, supporting our visual observation. We examined, pairwise, the performance of $Fe^{2+}$ vs. $Al^{3+}$, $MgCl_2$ vs. $CaCl_2$, and $MgSO_4$ vs. $CaSO_4$, and neither compared cations performed better than their counterparts (S1 Appendix: two sample t-test with equal variances, t = 0.20, 0.33 and -0.04, respectively, where t_crit = 2.30).

## Discussion

The efficiency of sedimentation ponds depends on a multitude of factors. The sediment amount and type it receives and the patterns how the pond is loaded are driven by the soil type upstream, in-field or edge-of-field measures, and climatic factors. The same climatic factors

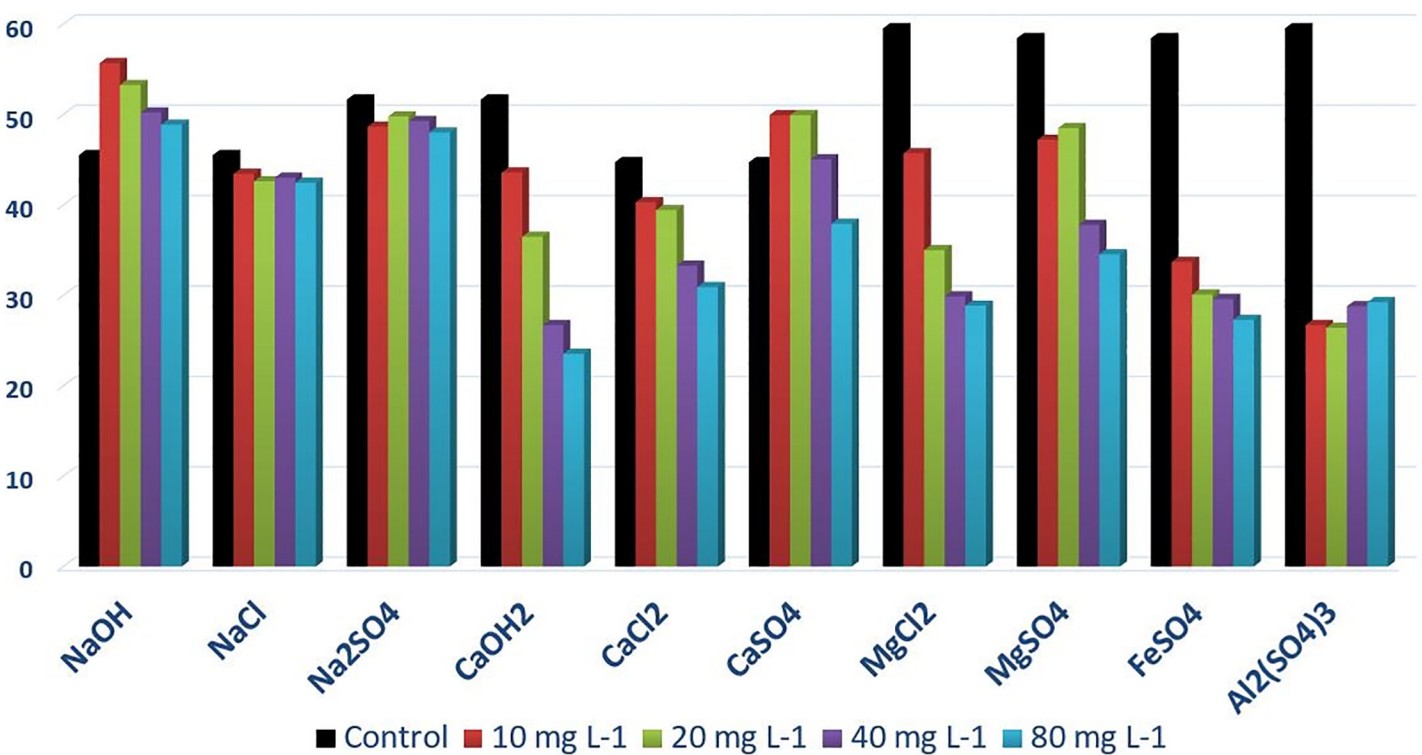

**Fig 6. Half-times of overall sedimentation–defined as the time required to achieve 50% of the expected final sedimentation efficiency, using an exponential model fitted to the data—using the cation-anion pairs introduced in Table 1 as coagulants.**

**Table 3. Chloride and sulphate effects on the sedimentation efficiency (%): three repetitions and their averages (mean) with the standard deviation (SD).**

| Removal efficiency, % | | 0 mg L (K) | 10 mg L$^{-1}$ | 20 mg L$^{-1}$ | 40 mg L$^{-1}$ | 80 mgL$^{-1}$ |
|---|---|---|---|---|---|---|
| MgCl$_2$ | 1 | 39.8 | 58.8 | 85.5 | 86.6 | 88.2 |
| | 2 | 40.1 | 58.3 | 87.1 | 88.9 | 90.6 |
| | 3 | 39.3 | 56.5 | 86.4 | 90.5 | 91.5 |
| | **mean (SD)** | **39.7 (0.3)** | **57.9 (1.0)** | **86.4 (0.7)** | **88.7 (1.6)** | **90.1 (1.4)** |
| CaCl$_2$ | 1 | 41.0 | 51.4 | 73.8 | 85.8 | 89.3 |
| | 2 | 40.1 | 52.5 | 75.9 | 87.5 | 90.9 |
| | 3 | 39.3 | 51.2 | 75.7 | 85.7 | 88.9 |
| | **mean (SD)** | **40.1 (0.7)** | **51.7 (0.6)** | **75.1 (1.0)** | **86.3 (0.8)** | **89.7 (0.9)** |
| MgSO$_4$ | 1 | 38.2 | 50.5 | 75.1 | 85.3 | 88.1 |
| | 2 | 41.3 | 51.8 | 74.9 | 87.6 | 91.4 |
| | 3 | 42.1 | 55.7 | 78.3 | 85.7 | 88.9 |
| | **mean (SD)** | **40.5 (1.7)** | **52.7 (2.2)** | **76.1 (1.6)** | **86.2 (1.0)** | **89.5 (1.4)** |
| CaSO$_4$ | 1 | 41.0 | 61.6 | 61.6 | 83.5 | 90.4 |
| | 2 | 41.3 | 50.3 | 65.5 | 85.2 | 91.5 |
| | 3 | 42.1 | 50.2 | 67.9 | 83.4 | 88.2 |
| | **mean (SD)** | **41.5 (0.5)** | **54.0 (5.4)** | **65.0 (2.6)** | **84.1 (0.8)** | **90.0 (1.3)** |

and the ponds' dimensioning will determine the ponds' hydraulic loading under normal and extreme conditions. The size of a pond can be constrained by natural or economic considerations, and once they are established, they are unlikely to be re-shaped. We therefore assumed that it is easier to look into the speed of sedimentation as point of influence, as from a technical point of view that is the easiest to influence and keep under control.

The mechanism behind artificially increasing sedimentation is coagulation–technology used in closed environments like that of waste-water treatment. Coagulation will bind suspended particles together with a certain force, and as a result those will present a larger effective diameter and thus settle faster, providing a technological opportunity. Different types of coagulants are known in industry, however, many of those are under strict environmental control, and their application in an open environment would be problematic. Materials that affect coagulation and are found in the natural environment are e.g. manyfold of salts. They are natural, some are used in various open-enviroment applications–such as road-salting, chemical improvers to certain kinds of soils, pH regulator to water bodies, nutrient (meso-element) supplies to plants and animals–and many are part of human diet as well, both naturally and in the form of dietary supplements. Therefore, we found it reasonable to establish a sequence of studies to examine their practical applicability in improving sedimentation efficiency in sedimentation ponds from the technical, environmental as well as economic point of view.

### Cation-anion effects on the rate and overall efficiency of sedimentation

As a first experiment, the progression of particle-sedimentation in salt-application treatments in static glass cylinders was quantified and evaluated using time-lapse photography and subsequent image analysis, in order to demonstrate their potential and pre-select some salts for further study. The approach included quantification of the RGB-sum of images taken of growingly transparent soil solutions as an increasing amount of solid particles sedimented over time. The approach returned logical results that were generally in line with applicable physico-chemical rules and laws (c.f. Stokes-law, Shulze-Hardy rule and the Brownian motion of colloides).

In terms of sedimentation efficiency, cation performance was very consistent with the Shulze-Hardy rule. We did not have the setup or the breadth of the study to examine the applicability of the power-law aspect of the rule, but ion charge was a good predictor of ion-performance, with little, insignificant differences among cations of the same charge, but a substantial differences between the performance of cations of different charges. Our data also cannot rank between the examined anions, as their performance was very close to each other. This was not surprising, given the contradictory reports about them in the literature.

Only one anion, $OH^-$ influenced pH strongly (shown in Fig 3B). Of the cations, $Na^+$ tended to increase the pH, while $Fe^{2+}$ and $Al^{3+}$ made the solution more acidic. Other ions did not make any noticeable and accountable impact on the pH of the solution. We note, however, that during a practical application in a sedimentation pond the salt would be diluted downstream, and the effect on pH would likely be less expressed. We avoid working with $OH^-$ further in follow-up experiments but note that in case lowering of pH would be desirable in a particular water-body, $OH^-$ can provide a potential secondary effect on pH, while the applicable cation may help sedimentation. However, it is to be accounted for that the strong alkalic pH by the $OH^-$ ions promote a dispersion effect, which works against the coagulation and eventual sedimentation.

The context for examining the speed of sedimentation is the fact that hydrological retention time–the time it takes to entirely replace the volume of water in a pond–may become substantially shorter during high-flow events, and therefore the rate of the coagulation/sedimentation effect may become important. According to the literature, sedimentation-wetland systems in

Norway are designed to have on average 4–4.5 hours of hydrological retention time under normal flow conditions. According to [29], 3 hours of retention time refers to intensive runoff conditions, which helped establish the duration of our lab experiments. It is to be understood that turbulence gets much more significant in high-flow conditions–something we have not yet tested for in the laboratory–and technology is needed to reduce that if enhanced sedimentation is to be achieved. These systems are also rather ineffective if precipitation and flood extremes occur and they get overflown.

## An optical approach to evaluate sedimentation efficiency in the laboratory

The obtained optical results were not calibrated to traditional laboratory measurements on how much of the applied solid particles settled out of the solution. We expect that results we obtained are likely specific to the kind of soil material, and such relationships between the optical and laboratory results may not necessarily be possible to generalize for other soil materials. At this stage in the broader study, we found it sufficient and more feasible to use the simplified and cheaper optical approach to examine the solutions' turbidity. One possible weakness of not cross-checking optical and laboratory results is that the assumption of linearity in the RGB-response within the defined 0–100% efficiency range may not be accurate. In our judgement this was acceptable given the engineered setup using only one soil material, and given that the goal of the study was not to introduce and evaluate the imaging method itself.

The impact of variation was quantified and/or ruled out at several stages of the study. We set up the experiment such that each run had its own control, conducted the image analysis such that any variation in light conditions was accounted for in both the vertical and horizontal dimension and discounted from the treatment-images, and we ran and evaluated three replications of select treatments. Replicability of the experiment for the salt-treatments appeared to be very good, as our data shows. At the same time, the control treatments were supposed to yield identical results across the entire experiment, whereas they varied to a certain degree, especially in terms of sedimentation half-time and the pH of the available water. We deem this partly to a combination of experimental variability and any slight day-to-day variation in the quality of the water and temperature–which may be magnified when salts are not present to dominate the sedimentation process. We also note that the exponential type curve is rather sensitive to the steep, initial parts in the data. To a lesser degree, this may also have affected the curves of the various treatments. For this reason we recommend putting more emphasis on trends and relative values rather than the presented absolute values of 'half-time'. Absolute values are expected to differ more from material to material as well.

## Aspects of potential environmental impact

The addition of salts to natural environments is generally undesirable. However, in some parts of the world, especially in places with a positive water balance, they are already in wide use for other applications–often in much greater dosages locally. In Norway, for instance Ca and Mg salts are among those used extensively in e.g. road salting (ca. 250,000 t/yr nationally [30]), ski-track preparation [31], road dust suppression [32], [33], i.e. they are not new to the Norwegian environment. At the same time their application has also been known and reported as nutrient supply to water-based ecosystems (especially to invertebrates [34]), or as soil amendments to improve soil pH and physico-chemical properties [35], [36]. Their artificial use often generates discussions, but the amounts used in such applications–especially in road salting—can be far greater than what is expected to be effective in the potential application that we envisage.

Due to the sustained problem of soil erosion and the apparent limitations to new phosphorus resources world-wide [37], we are under pressure to develop techniques that help keep

those valuable resources reusable for man before they reach the ocean-floor and become effectively unusable. We propose that using salt(s) in an intermittent fashion (i.e. timed pulses) and in generally low dosage to water directly may be generally justifiable if its environmental impact is properly studied, and a positive environmental and economic impact in retaining sediments and bound phosphorus is seen in return as a trade-off. Locations of applicability need to be considered carefully, but in many cases the desirable application sites may be close to sea, where the impacted water system downstream is very short. The proposed concentrations for intermittent use of salt-coagulants are well under those found harmful for sensitive aquatic species [38], and would get further diluted downstream towards background ion concentrations found in Norwegian surface waters (EEA database; Lakes in Norway, 1990–2012 [39]: Chloride ($Cl^-$): 0.5–4.5 $mgL^{-1}$, Sulfate ($SO_4^{2-}$): 0.6–4.5 $mgL^{-1}$, Magnesium ($Mg^{2+}$): 0.3–1.0 $mgL^{-1}$, Calcium ($Ca^{2+}$): 0.5–7.2 $mgL^{-1}$, Natrium ($Na^+$): 0.5–4.0 $mgL^{-1}$, pH: 5.0–7.8).

To our understanding, this study reported novel results in that it quantified the expected effects of varied levels of easily available salt-type coagulants on the sedimentation of non-ideal soil material; referring to natural soil that has not been cleared of organic and other binding substances. The soil type used is of a fine textured marine sediment that is common and is prone to erosion in SE Norway. In this respect the study links to a future field-test site. Our study also helped rule out some salt-based coagulants that may raise environmental concerns if applied in the field.

## Conclusions

We performed a limited-scope static experiment on salt-enhanced sedimentation of soil material in aqueous solution, in order to mimic the sedimentation process in still water, approximating conditions in a sedimentation pond system with low flow and turbulence. We found that salts of bivalent cations, such as $MgSO_4$, $MgCl_2$, $CaSO_4$ and $CaCl_2$ could increase the efficiency of sedimentation from ca. 40% to the range of 80–90% when applied in 40 mg $L^{-1}$ cation equivalent, but chlorides were nearly as effective at even half that dosage.

We see this as a first step towards a broader in-situ experiment with select salts as coagulants that may be acceptable in a real-world test and potential application. In a follow up experiment, we are checking the chemical and biological response of the natural system to such mild salt coagulant application downstream. It is also understood that innovation may be necessary in pond design to respond to both low and higher flow situations more optimally in terms of hydraulic retention time and turbulence in the system. We also plan to evaluate the environmental and economic impact of improved sediment and nutrient retention—and potential re-use—at the farm-scale, via valuating soil as a natural resource.

## Supporting information

**S1 Appendix. Statistical analysis to prove removal efficiency differences between the cations and anions.**
(XLSX)

## Author Contributions

**Conceptualization:** Andras Sebok, Imre Czinkota, Attila Nemes.

**Data curation:** Andras Sebok, Imre Czinkota, Attila Nemes.

**Formal analysis:** Andras Sebok, Imre Czinkota.

**Funding acquisition:** Attila Nemes.

**Investigation:** Andras Sebok, Viktoria Labancz.

**Methodology:** Andras Sebok, Imre Czinkota, Attila Nemes.

**Project administration:** Attila Nemes.

**Resources:** Andras Sebok, Viktoria Labancz, Attila Nemes.

**Software:** Imre Czinkota.

**Validation:** Andras Sebok, Attila Nemes.

**Visualization:** Andras Sebok, Attila Nemes.

**Writing – original draft:** Andras Sebok, Attila Nemes.

**Writing – review & editing:** Andras Sebok, Attila Nemes.

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
