## [Decision Letter · Decision Letter 0]

25 Jul 2019

PONE-D-19-17683

The effect of various metal-salts on the sedimentation of soil in a water-based suspension

PLOS ONE

Dear Mr Sebők,

Thank you for submitting your manuscript to PLOS ONE. After careful consideration, we feel that it has merit but does not fully meet PLOS ONE’s publication criteria as it currently stands. Therefore, we invite you to submit a revised version of the manuscript that addresses the points raised during the review process.

We would appreciate receiving your revised manuscript by Sep 08 2019 11:59PM. To enhance the reproducibility of your results, we recommend that if applicable you deposit your laboratory protocols in protocols.io, where a protocol can be assigned its own identifier (DOI) such that it can be cited independently in the future. For instructions see: http://journals.plos.org/plosone/s/submission-guidelines#loc-laboratory-protocols

We look forward to receiving your revised manuscript.

Kind regards,

Debarati Bhaduri, Ph.D.

Academic Editor

PLOS ONE

Journal Requirements:

1. In your Methods section, please including geographic coordinates for the data set if available.

3. Please ensure that you refer to Figure 6 in your text as, if accepted, production will need this reference to link the reader to the figure.

Additional Editor Comments (if provided):

Authors need to work on further in this manuscript to make it in a desired shape.

Both reviewers are partially satisfied and suggested number of changes for the clarity and betterment of the manuscript.

Hence authors are suggested to address the issues raised by the reviewers for further consideration in journal.

Reviewers' comments:

Reviewer's Responses to Questions

**Comments to the Author**

1. Is the manuscript technically sound, and do the data support the conclusions?

Reviewer #1: Partly

Reviewer #2: Partly

2. Has the statistical analysis been performed appropriately and rigorously? 

Reviewer #1: No

Reviewer #2: No

3. Have the authors made all data underlying the findings in their manuscript fully available?

Reviewer #1: Yes

Reviewer #2: Yes

4. Is the manuscript presented in an intelligible fashion and written in standard English?

Reviewer #1: Yes

Reviewer #2: Yes

5. Review Comments to the Author

Reviewer #1: Review report on “The effect of various metal-salts on the sedimentation of soil in a water-based suspension” (PONE-D-19-17683)

The present article evaluated the efficiency of different combinations of cations and anions at different concentrations for speedy sedimentation of suspended soil particles under static condition with a goal to upscale it and employ it in sedimentation ponds to reduce soil and associated nutrient loss to water bodies after erosion. In general, the manuscript is well written. Authors have also very aptly pointed out the limitations and boundary conditions of the present study. However, the manuscript has to be improved keeping in mind the following comments before it can be accepted for publication:

Major comments

1) Line 57-59: “Nutrients tend to bind stronger to these smaller particles due to their greater relative surface area and by that their greater adsorption capacity ([8], [9]).”- I think the greater adsorption capacity of small sized (< 2 µm) soil particles is mainly due to their charge characteristics, rather than the size itself. In the entire manuscript, the main focus has been on size of particles, but sedimentation of colloidal particles due to addition of excess cations or anions is due to suppression of diffuse double layer, a charge characteristic. This point should also be mentioned properly in ‘Introduction’ and ‘Discussion’ sections.

2) Line 114: “10, 20, 40 and 80 mg/L”- It is not clear whether these concentrations are with respect to the soil suspension or the solution of salts added? If these are salt solution concentrations then how much mL of salt solution was added to 500 mL suspension?

3) Table 1: NaOH and Ca(OH)2 are not salts, they are strong alkalis. These two are supposed to increase the pH of suspension, and consequently the pH dependent negative charges on colloidal particles. What was the logic behind using these two? How much was the pH increase due to addition of these two?

4) Line 169-173: Please explicitly mention the equation as well show how ‘t’ times ‘ln2’ gives ‘half-time’ to make it more intelligible.

5) Table 3: Authors could consider to run a two-way ANOVA taking ‘type of salt’ and ‘concentration of salt’ as two factors to show their individual effect on sedimentation as well as to identify the most appropriate concentration of the most suitable salt for achieving required sedimentation efficiency. In that case, there is no need to provide replication wise data; mean values with standard error of mean along with group letters based on Fisher’s LSD or Tukey’s HSD would be fine. Accordingly one sentence from this table should be added to ‘Conclusions’.

6) Line 287-292: What about the dispersion that is likely to be promoted by addition of a strong alkali like NaOH?

Minor comments

1) Line 46-48: “The ponds’ efficiency varies greatly, however, and depends on many factors, among them the sediment type, the pond’s dimensions and its nominal hydraulic loading ([4], [5]).”- Re-write this sentence to make it more meaningful. I think authors wanted to point out that factors like “sediment type, the pond’s dimensions and its nominal hydraulic loading” are more important than others.

2) Line 100: It should be 13.9% instead of 13,9%

3) Please use ‘L’ instead of ‘l’ to indicate ‘Litre’ throughout the manuscript

4) Line 186: It should be ‘comprehensible’ instead of ‘comprehendible’

5) Table 2: Please correct the “ before half-time

6) Line 331: Is it 250.000 or 250,000? Please check

7) Line 351-352: Please correct the values. There are commas instead of decimal points.

Reviewer #2: Although the MS is reviewed for a very small number of days. I have certain queries that has to be resolved before it is finally accepted in PLOS One.

1. Line 45-46- Reports ...2 and 3. Reframe the sentence for clarity.

2. Line 47-48- The ponds efficiency...4,5. Kindly reframe it as “among them are”.

3. Line 57- It is Brownian motion. So change it accordingly across the MS.

4. Line 74-76- Under reductive condition...disappear over time. The statement is not clear. Kindly elaborate what are the bonds you want to explain here. Secondly what is the redox potential at this point.

5. Reference number 20 and 21 seems too old here presented as a basis for taking experiment.

6. Line 82- working hypothesis is to enhance particle coagulation. I assume this can be extrapolated to aquatic systems. But, I am unable to find any solid reasons to justify as the soil and aquatic systems are totally different.

7. Line 100- correct 13,9%.

8.Line 102- correct the spelling of course

9.Line 105- explain the deviation observed in the protocol

10. The statistical methods used for the experiment needs to be clear enough like see the presented data in table and figures are without SD and SE

11. Line 214 to 216 needs a recent reference.

12. The discussion section just orients towards the science of coagulation with Monovalent and Divalent ions. Hence the authors must present novelty in this work.

13. They can discuss the ions individually. Please check the new references in that direction.

14. Line 286- OH ions influences the pH. The authors should have presented the data for clarity. Speculations in the discussion section is partly acceptable.

15. The reference section is not uniform. See Blankenberg et al and Sveistrup et al.

I hope to see these changes before final publication.

6. PLOS authors have the option to publish the peer review history of their article (what does this mean?). If published, this will include your full peer review and any attached files.

Reviewer #1: No

Reviewer #2: No

---

## [Author Response · Author response to Decision Letter 0]

11 Oct 2019

We attached a "cover letter" with the answers for the reviewers.

---

## [Decision Letter · Decision Letter 1]

20 Nov 2019

PONE-D-19-17683R1

The effect of various metal-salts on the sedimentation of soil in a water-based suspension

PLOS ONE

Dear Mr Sebők,

Thank you for submitting your manuscript to PLOS ONE. After careful consideration, we feel that it has merit but does not fully meet PLOS ONE’s publication criteria as it currently stands. Therefore, we invite you to submit a revised version of the manuscript that addresses the points raised during the review process.

We would appreciate receiving your revised manuscript by Jan 04 2020 11:59PM. To enhance the reproducibility of your results, we recommend that if applicable you deposit your laboratory protocols in protocols.io, where a protocol can be assigned its own identifier (DOI) such that it can be cited independently in the future. For instructions see: http://journals.plos.org/plosone/s/submission-guidelines#loc-laboratory-protocols

We look forward to receiving your revised manuscript.

Kind regards,

Debarati Bhaduri, Ph.D.

Academic Editor

PLOS ONE

Additional Editor Comments (if provided):

Minor corrections:

1. Abstract (& elsewhere): ml should be corrected to mL

2. Abstract (& elsewhere): mg/L should be written as mg L-1

3. Table 1: Give justification of 'Not Tested' cation and anion combinations

4. Table 3: 'avg' is not a standard term, should be replaced with average or mean

5. Abstract L10-17: should not be such elaborated, reduce this part.

6. Discussion: As this section is long, try to divide with 2-3 suitable sub-headings.

Reviewers' comments:

Reviewer's Responses to Questions

**Comments to the Author**

1. If the authors have adequately addressed your comments raised in a previous round of review and you feel that this manuscript is now acceptable for publication, you may indicate that here to bypass the “Comments to the Author” section, enter your conflict of interest statement in the “Confidential to Editor” section, and submit your "Accept" recommendation.

Reviewer #1: All comments have been addressed

Reviewer #2: All comments have been addressed

2. Is the manuscript technically sound, and do the data support the conclusions?

Reviewer #1: Yes

Reviewer #2: Yes

3. Has the statistical analysis been performed appropriately and rigorously? 

Reviewer #1: Yes

Reviewer #2: Yes

4. Have the authors made all data underlying the findings in their manuscript fully available?

Reviewer #1: Yes

Reviewer #2: Yes

5. Is the manuscript presented in an intelligible fashion and written in standard English?

Reviewer #1: Yes

Reviewer #2: Yes

6. Review Comments to the Author

Reviewer #1: All the questions raised by me have been answered. The manuscript in its current form can be accepted.

Reviewer #2: All the points are now well answered and incorporated. The editors may consider it for publication.

7. PLOS authors have the option to publish the peer review history of their article (what does this mean?). If published, this will include your full peer review and any attached files.

Reviewer #1: No

Reviewer #2: No

---

## [Author Response · Author response to Decision Letter 1]

11 Dec 2019

Dear Editor,

We appreciate your work and your comments to improve our article. We addressed all of your suggested modifications:

Comment 1&2: We changed all of the units to the form you marked.

Comment 3: In the description (L124-126) we add a sentence to clarify, why we did not test the marked salts.

Comment 4: The 'avg' expression changed to 'mean'.

Comment 5: We decreased the abstract and modified some sentence at L 10-17.

Comment 6: We divided the discussion into 3 smaller parts wit sub-headings.

We hope that these changes answer all of your questions.

Kind regards: The authors.

---

## [Editor Report · Decision Letter 2]

18 Dec 2019

The effect of various metal-salts on the sedimentation of soil in a water-based suspension

PONE-D-19-17683R2

Dear Dr. Sebők,

We are pleased to inform you that your manuscript has been judged scientifically suitable for publication and will be formally accepted for publication once it complies with all outstanding technical requirements.

With kind regards,

Debarati Bhaduri, Ph.D.

Academic Editor

PLOS ONE

Additional Editor Comments (optional):

Authors have addressed all comments with due care.
---

## [Editor Report · Acceptance letter]

23 Dec 2019

PONE-D-19-17683R2 

The effect of various metal-salts on the sedimentation of soil in a water-based suspension 

Dear Dr. Sebok:

I am pleased to inform you that your manuscript has been deemed suitable for publication in PLOS ONE. Congratulations! Your manuscript is now with our production department. 

With kind regards,

on behalf of

Dr. Debarati Bhaduri 

Academic Editor

PLOS ONE